# HELZ directly interacts with CCR4–NOT and causes decay of bound mRNAs

Aoife Hanet[1], Felix Räsch[1], Ramona Weber[1], Vincenzo Ruscica[1], Maria Fauser[1], Tobias Raisch[1,2], Duygu Kuzuoğlu-Öztürk[1,3], Chung-Te Chang[1], Dipankar Bhandari[1], Cátia Igreja[1], Lara Wohlbold[1]

Eukaryotic superfamily (SF) 1 helicases have been implicated in various aspects of RNA metabolism, including transcription, processing, translation, and degradation. Nevertheless, until now, most human SF1 helicases remain poorly understood. Here, we have functionally and biochemically characterized the role of a putative SF1 helicase termed "helicase with zinc-finger," or HELZ. We discovered that HELZ associates with various mRNA decay factors, including components of the carbon catabolite repressor 4–negative on TATA box (CCR4–NOT) deadenylase complex in human and *Drosophila melanogaster* cells. The interaction between HELZ and the CCR4–NOT complex is direct and mediated by extended low-complexity regions in the C-terminal part of the protein. We further reveal that HELZ requires the deadenylase complex to mediate translational repression and decapping-dependent mRNA decay. Finally, transcriptome-wide analysis of *Helz*-null cells suggests that HELZ has a role in the regulation of the expression of genes associated with the development of the nervous system.

## Introduction

RNA helicases are ubiquitous enzymes that mediate ATP-dependent unwinding of RNA duplexes and promote structural rearrangements of RNP complexes. They participate in all aspects of RNA metabolism such as transcription, processing, translation, ribosome assembly, and mRNA decay (Bleichert & Baserga, 2007). There are six helicase superfamilies (SFs) 1–6 defined by sequence, structure, and mechanism (Singleton et al, 2007). Eukaryotic helicases belong exclusively to either SF1 or SF2, which are characterized by a structural core composed of tandem RecA-like domains and as many as 12 conserved sequence motifs that mediate substrate binding, catalysis, and unwinding (Fairman-Williams et al, 2010). Approximately 70 RNA helicases are known to be expressed in human cells, most of which belong to the SF2 superfamily, such as the well characterized DEAD (Asp-Glu-Ala-Asp)-box family of helicases (Sloan & Bohnsack, 2018). To date, only 11

SF1 RNA helicases have been identified; among them is the highly conserved upstream frameshift 1 (UPF1) helicase, which has an important role in nonsense-mediated mRNA decay (Kim & Maquat, 2019). Few other eukaryotic UPF1-like SF1 helicases have been investigated in detail. Senataxin, the human orthologue of yeast Sen1p, has a role in transcriptional regulation (Ursic et al, 2004; Chen et al, 2006; Leonaite et al, 2017). Other examples are the mammalian moloney leukemia virus homolog 10 (MOV10), the fly Armitage and the silencing defective protein 3 (SDE3) in plants, which all function in post-transcriptional gene silencing (Dalmay et al, 2001; Cook et al, 2004; Burdick et al, 2010; Gregersen et al, 2014).

The putative RNA "Helicase with Zinc-finger" (HELZ) is conserved in Metazoa and belongs to the UPF1-like family of SF1 helicases (Fairman-Williams et al, 2010). The gene encoding HELZ was cloned from a human immature myeloid cell line cDNA library (KIA0054) over 20 years ago but its cellular function remains poorly studied (Nomura et al, 1994). HELZ helicases are large proteins that contain a $Cys_3His$ (CCCH)-type zinc finger (ZnF) motif N-terminal to the helicase core, and a largely unstructured C-terminal half with a conserved polyadenosine (poly[A])-binding protein (PABP)–interacting motif 2 (PAM2) (Fig 1A). The C-terminal half of HELZ varies in size and sequence depending on the species. Two LxxLAP (Leu, x indicates any amino acid, Leu, Ala, Pro) motifs are also embedded within the HELZ C-terminal region; these motifs are found in hypoxia-inducible transcription factors and regulate their stability in response to oxygen depletion; HELZ abundance, however, does not appear to be associated with oxygen levels (Hasgall et al, 2011).

Murine HELZ has a widespread spatial and temporal expression throughout embryonic development (Wagner et al, 1999). Human HELZ is a component of complexes containing the RNA Polymerase II, as well as the histone methyltransferases Smyd2 or Smyd3, which indicates a target-specific role in transcription (Hamamoto et al, 2004; Diehl et al, 2010). HELZ stimulates translation when overexpressed in human cells and interacts with cytoplasmic polyadenylate-binding protein 1 (PABPC1) (Hasgall et al, 2011). PABPs represent a major class of mRNA-regulating proteins that interact with the poly(A) tail of mRNAs, thereby influencing their stability and translation efficiency (Goss & Kleiman, 2013; Nicholson & Pasquinelli, 2019). The shortening of the poly(A) tail

[1]Department of Biochemistry, Max Planck Institute for Developmental Biology, Tübingen, Germany  [2]Department of Structural Biochemistry, Max Planck Institute of Molecular Physiology, Dortmund, Germany  [3]Helen Diller Family Cancer Research, University of California San Francisco, San Francisco, CA, USA

Correspondence: catia.igreja@tuebingen.mpg.de; lara.wohlbold@tuebingen.mpg.de

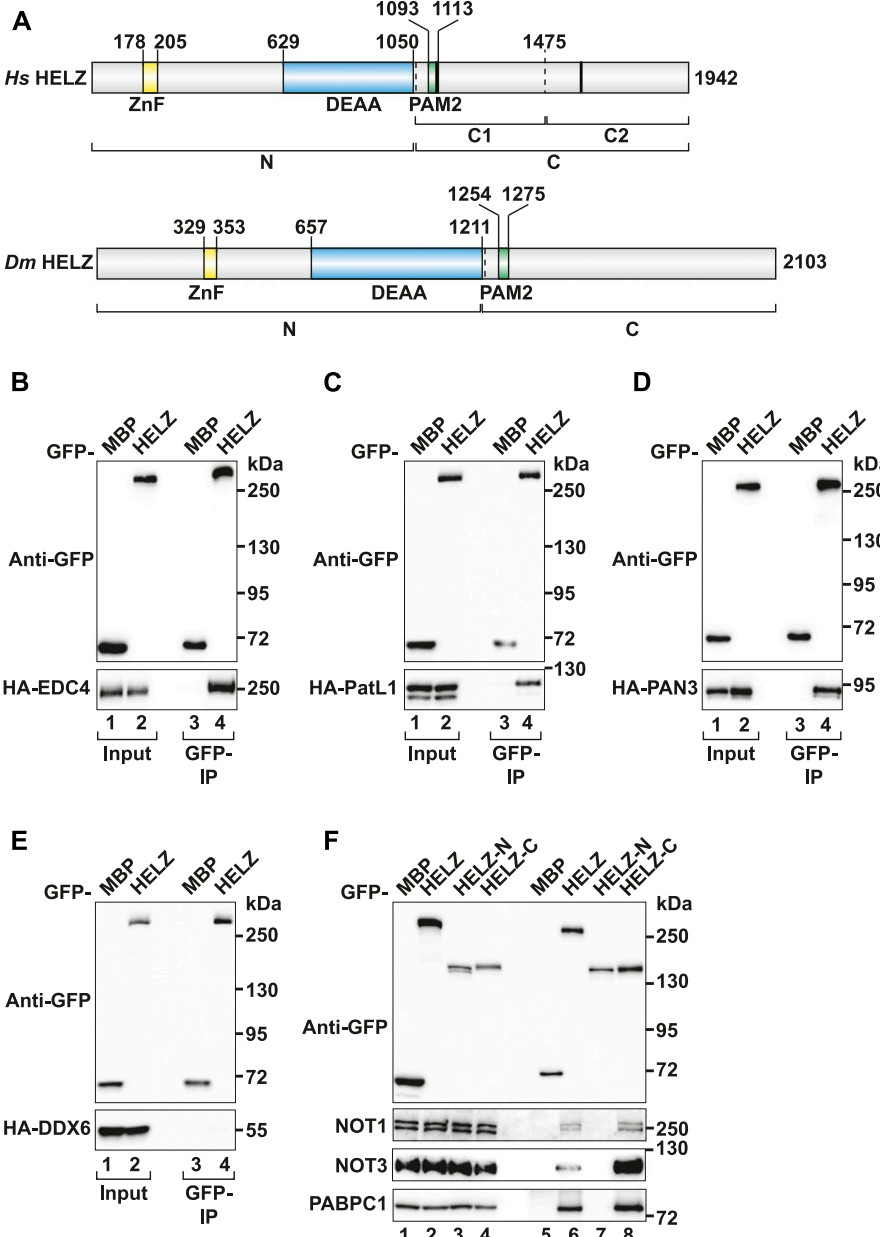

**Figure 1.   HELZ interacts with mRNA decay factors.**
**(A)** Schematic representation of *Hs* HELZ and *Dm*
HELZ. The Zinc finger (ZnF), the putative helicase (DEAA,
Asp, Glu, Ala, Ala) domain, and the PABP interacting
motif 2 (PAM2) are highlighted in yellow, blue, and
green, respectively. Black bars indicate the position of
the previously described LxxLAP motifs in *Hs* HELZ
(Hasgall et al, 2011). HELZ N- and C-terminal
fragments are indicated below the scheme. Border
residue numbers are listed above the scheme.
**(B–E)** Immunoprecipitation assay in HEK293T cells
showing the interaction of GFP-HELZ with HA-tagged
EDC4 (B), HA-tagged-PatL1 (C), HA-tagged-PAN3 (D), and
HA-tagged-DDX6 (E). GFP-MBP served as negative
control. Input (2% for GFP-tagged proteins and 1%
for HA-tagged proteins) and bound fractions (20% for
GFP-tagged proteins and 30% for HA-tagged
proteins) were analysed by Western blotting. **(F)**
Immunoprecipitation assay in HEK293T cells showing
the interaction of GFP-tagged HELZ (full-length and
indicated fragments) with endogenous NOT1, NOT3,
and PABPC1. Input (1.2%) and bound fractions (20% for
GFP-tagged proteins and 35% for endogenous
proteins) were analysed by Western blotting.
Source data are available for this figure.

and concomitant release of PABPC1, a process termed deadenylation, is a critical determinant of mRNA stability and translational efficiency (Inada & Makino, 2014; Webster et al, 2018). HELZ was detected in a screen for helicases that interact with the carbon catabolite repressor 4-negative on TATA box (CCR4–NOT) complex (Mathys et al, 2014), the major cytoplasmic deadenylase in eukaryotes (Yi et al, 2018). The association of HELZ with the deadenylase complex hints at an important but presently uncharacterized role of this helicase in regulating stability and translation of mRNA.

In this study, we show that human HELZ directly interacts with the NOT module of the CCR4–NOT complex via multiple motifs embedded within the low-complexity region of the protein. In tethering assays with reporter mRNAs, HELZ elicits deadenylation

followed by decapping and subsequent 5'-to-3' exonucleolytic decay. The ability of HELZ to induce decay of bound mRNAs is conserved in Metazoa and depends on the CCR4–NOT complex. We also provide evidence that tethered HELZ can repress translation independently of mRNA decay in a manner dependent on both the CCR4–NOT complex and the DEAD-box helicase DDX6. Finally, using transcriptome sequencing, we identified 3,512 transcripts differentially expressed (false discovery rate [FDR] < 0.005) in *Helz*-null cells. Interestingly, many of the up-regulated mRNAs are linked with the development of the nervous system.

Taken together, our data reveal an important function of HELZ in governing the expression of specific genes, possibly through both transcriptional and posttranscriptional regulatory mechanisms.

# Results

### HELZ interacts with mRNA decay factors

HELZ is a largely uncharacterized protein implicated in post-transcriptional gene regulation (Hasgall et al, 2011; Mathys et al, 2014). To identify novel HELZ-interacting partners, we performed co-immunoprecipitation (co-IP) assays using overexpressed GFP-tagged *Hs* HELZ as bait against different hemagglutinin (HA)-tagged proteins in human HEK293T cells. HELZ interacted with multiple mRNA decay factors, including the decapping enhancers EDC4 and PatL1 as well as the poly(A) specific ribonuclease subunit 3 (PAN3) subunit of the PAN2/PAN3 deadenylase complex (Fig 1B–D). However, under the co-IP conditions, we did not detect an interaction with DDX6, as previously identified by mass spectrometry (Ayache et al, 2015) (Fig 1E). GFP-HELZ readily immunoprecipitated the endogenous CCR4–NOT deadenylase complex proteins NOT1 and NOT3 (Fig 1F, lane 6), suggesting that HELZ associates with the fully assembled complex in cells. PABPC1, which binds to HELZ via its PAM2 motif (Hasgall et al, 2011), was used as a positive control.

To delineate the region of HELZ critical for the interaction with CCR4–NOT, we divided the HELZ protein into an N-terminal fragment encompassing the ZnF motif and the helicase domain (HELZ-N, Table S1) and a second fragment comprising the low-complexity C-terminal region of HELZ including the PAM2 motif (HELZ-C, Table S1 and Fig 1A). Both fragments were then tested separately for their ability to interact with NOT1 and NOT3. Interestingly, the HELZ-C fragment was sufficient to mediate binding to NOT1 and NOT3 as well as PABPC1. In contrast, HELZ-N did not interact with any of these proteins (Fig 1F, lanes 7 and 8).

### HELZ directly binds CCR4–NOT via multiple C-terminal sites

The CCR4–NOT complex consists of several subunits arranged around the scaffold protein NOT1 (Collart & Panasenko, 2017). NOT10 and NOT11 bind to the N-terminal region of NOT1 (Lau et al, 2009; Bawankar et al, 2013; Mauxion et al, 2013). The catalytically active nucleases CAF1 (or its paralog POP2) and CCR4a (or its paralog CCR4b) bind to a central MIF4G (middle-domain of eIF4G)-like domain of NOT1 (Lau et al, 2009; Basquin et al, 2012; Petit et al, 2012) adjacent to the CAF40-binding domain (CC) of NOT1 (Chen et al, 2014; Mathys et al, 2014). The CC domain is followed by a short connector domain in NOT1, recently identified to be an additional MIF4G-like domain, termed MIF4G-C (Raisch et al, 2018). NOT2 and NOT3 assemble on the C-terminal part of NOT1 (Bhaskar et al, 2013; Boland et al, 2013).

To test whether the interaction of HELZ with the CCR4–NOT complex is direct, we performed pull-down assays with recombinant and purified proteins. Production of intact HELZ-C in bacteria was not possible as it was very susceptible to proteolytic degradation. Therefore, we divided HELZ-C into two non-overlapping fragments of roughly similar size: HELZ-C1 and HELZ-C2 (Table S1 and Fig 1A). These fragments, fused to an N-terminal maltose-binding protein (MBP) and a C-terminal B1 domain of immunoglobulin-binding protein G (GB1)-hexahistidine tag (Cheng & Patel, 2004), were more stable during bacterial production. Following capture by nickel affinity, the eluted HELZ fragments were incubated with different recombinant human CCR4–NOT subcomplexes and pulled down via the MBP tag. In detail, we tested the interaction of HELZ with a pentameric subcomplex comprising a NOT1 fragment lacking the N-terminal region bound to CAF1, CAF40, and the C-terminal domains of NOT2 and NOT3 (Fig 2A) (Sgromo et al, 2017). HELZ-C1 and HELZ-C2 fragments both pulled down the pentameric subcomplex (Fig 2B and C, lanes 20). To elucidate which subunits of the pentameric subcomplex are involved in the interaction with HELZ, we also analysed binding to the CAF1/NOT1-MIF4G heterodimer, the CAF40 module (CAF40/NOT1-CC), the subsequent NOT1 MIF4G-C domain (CD), and the NOT module (NOT1/2/3) (Fig 2A) (Sgromo et al, 2017). HELZ-C1 and HELZ-C2 fragments both pulled down the NOT module of CCR4–NOT (Fig 2B and C, lane 24). Neither fragment interacted with the CAF1 module, the CAF40 module, or the MIF4G-C domain (Fig 2B and C, lanes 21–23). We conclude that human HELZ directly binds the NOT module using multiple sites in the low-complexity C-terminal region.

### HELZ induces 5′-to-3′ decay of tethered reporter mRNAs

To address the role of HELZ in the regulation of mRNA stability, we performed MS2-based tethering assays in HEK293T cells. We used a β-globin mRNA reporter containing six MS2-binding sites in the 3′ UTR (β-globin-6xMS2bs) and co-expressed full-length HELZ with an MS2-HA-tag (Fig 3A–C) (Lykke-Andersen et al, 2000). Tethering of HELZ resulted in a threefold reduction in the β-globin-6xMS2bs mRNA levels compared with the control protein MS2-HA (Fig 3A and B). The levels of a control reporter mRNA lacking the 6xMS2bs (control) were unaffected (Fig 3B). Consistent with the ability to bind CCR4–NOT, the C-terminal region of HELZ was sufficient to trigger mRNA decay when tethered to the same reporter mRNA. In contrast, the N-terminal region of HELZ containing the ZnF and helicase core did not induce decay of the reporter mRNA (Fig 3A and B).

We then tested whether HELZ binding to PABPC1 is required to induce decay of the tethered reporter mRNA. We introduced a point mutation in the HELZ PAM2 motif (F1107V) that specifically disrupts the interaction with PABPC1 (Fig 3D, lane 6) (Kozlov et al, 2001; Berlanga et al, 2006). Interestingly, the F1107V mutation did not alter the ability of HELZ to reduce the abundance of the bound mRNA reporter (Fig 3E–G), indicating that binding to PABPC1 is not required for this function.

To determine if a functional CCR4–NOT complex was necessary for HELZ-mediated degradation of bound mRNAs in cells, we first impaired the deadenylation activity of the CCR4–NOT complex by overexpressing a catalytically inactive mutant of CAF1 (CAF1*; D40A/E42A), which replaces the endogenous enzyme in a dominant negative manner (Horiuchi et al, 2009; Huntzinger et al, 2013). In addition, we overexpressed the Mid-region of NOT1 (residues T1085–T1605) to compete with endogenous NOT1 and sequester CAF1/CCR4 deadenylases as well as CAF40 from the endogenous deadenylase complex, compromising its activity. Overexpression of CAF1*/NOT1-Mid, together with MS2-HA-HELZ, led to a marked stabilization of the β-globin-6xMS2bs mRNA (Fig 3H–J). This is consistent with a model in which mRNA decay triggered by HELZ requires CCR4–NOT–mediated deadenylation.

We then blocked mRNA decapping by overexpressing a catalytically inactive mutant of DCP2 (DCP2*; E148Q) (Wang et al, 2002; Chang et al, 2014). This resulted in the accumulation of a fast

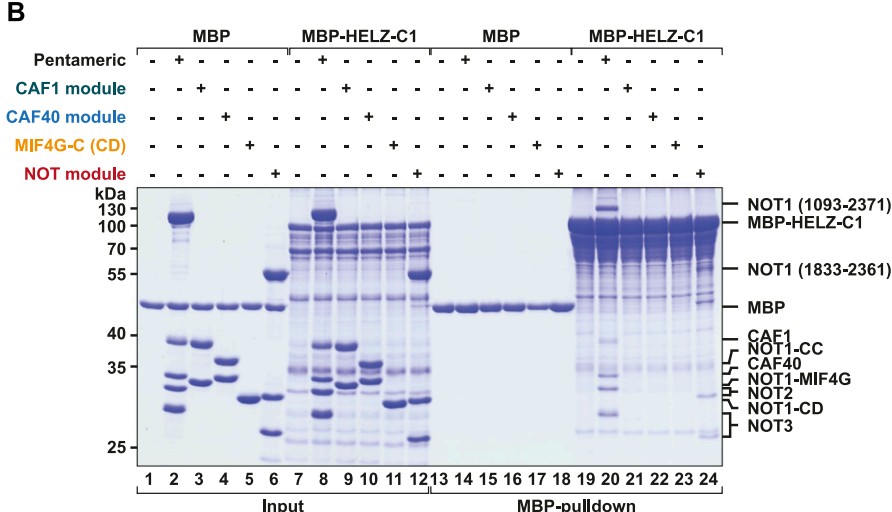

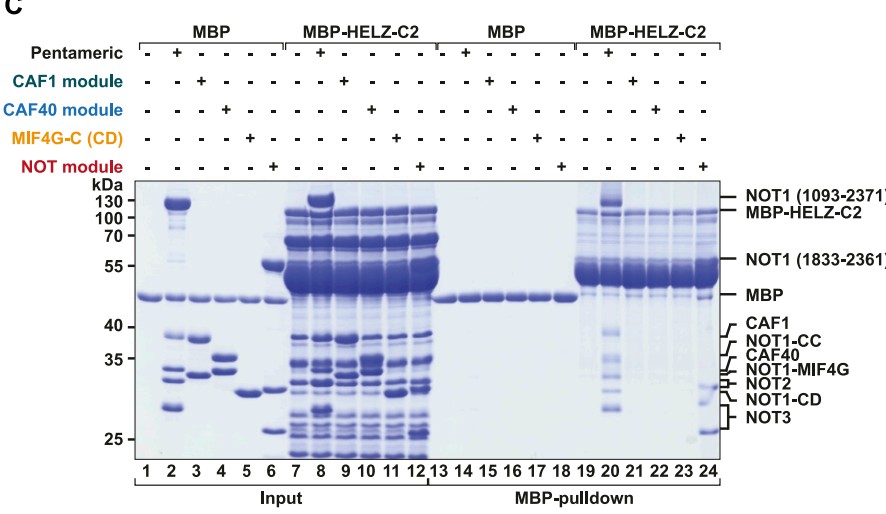

**Figure 2. HELZ directly binds CCR4–NOT via multiple C-terminal sites.**

**(A)** Schematic overview of the pentameric human CCR4–NOT complex used for in vitro interaction studies. The pentameric subcomplex is composed of NOT1 (residues E1093–E2371), CAF1, CAF40 (residues R19–E285), NOT2 (residues T344–F540), and NOT3 (residues L607–Q753). The CAF1 module contains the NOT1 MIF4G-like domain and CAF1 (green). The CAF40 module consists of CAF40 (blue; residues R19–E285) bound to the CAF40-binding coiled coil domain (CC; residues V1351–L1588). The adjacent NOT1 MIF4G-C (CD; residues D1607–S1815) is depicted in yellow. The NOT module consists of NOT1 (residues H1833–M2361), NOT2 (residues M350–F540; purple), and NOT3 (residues L607–E748; red). **(B, C)** In vitro MBP pull-down assay showing the interaction of recombinant MBP-*Hs* HELZ-C1-GB1-His (B) or MBP-*Hs* HELZ-C2-GB1-His (C) with distinct recombinant and purified CCR4–NOT modules (indicated on top of the respective gel). MBP served as a negative control. Input (33%) and eluted fractions (55%) were analysed by SDS–PAGE and Coomassie Blue staining.
Source data are available for this figure.

migrating reporter mRNA intermediate that lacks a poly(A) tail upon tethering of MS2-HA-HELZ to the β-globin-6xMS2bs reporter. MS2-HA-NOT1 served as a positive control for deadenylation-dependent mRNA decapping (Fig 3K–M) (Kuzuoglu-Ozturk et al, 2016). To confirm that this mRNA intermediate is indeed deadenylated, we performed an oligo(dT)-directed ribonuclease H (RNase H) cleavage assay. Poly(A) tail cleavage by RNase H of the reporter mRNAs (control and β-globin-6xMS2bs) in cells expressing MS2-HA and DCP2* resulted in the accumulation of fast migrating bands (Fig S1A, lane 1 versus 3; $A_n$ versus $A_0$). In contrast, in cells expressing MS2-HA-HELZ and DCP2*, the β-globin-6xMS2bs mRNA migrated as the deadenylated version of the reporter before and after the RNase H

treatment (Fig S1A, lane 2 versus 4). Based on these observations, we conclude that in human cells, HELZ promotes CCR4–NOT–dependent deadenylation followed by deadenylation-dependent degradation of the tethered mRNA.

### The role of HELZ in inducing mRNA decay is conserved in Metazoa

*Drosophila melanogaster (Dm)* HELZ, denoted as CG9425 (FlyBase/DIOPT: DRSC integrative orthologue prediction tool [Hu et al, 2011; Gramates et al, 2017]), displays a domain organization similar to that of *Hs* HELZ (Fig 1A) and shares an overall sequence identity of 31.38% (17.65% for the nonconserved C-terminal sequences) (UniProt Clustal

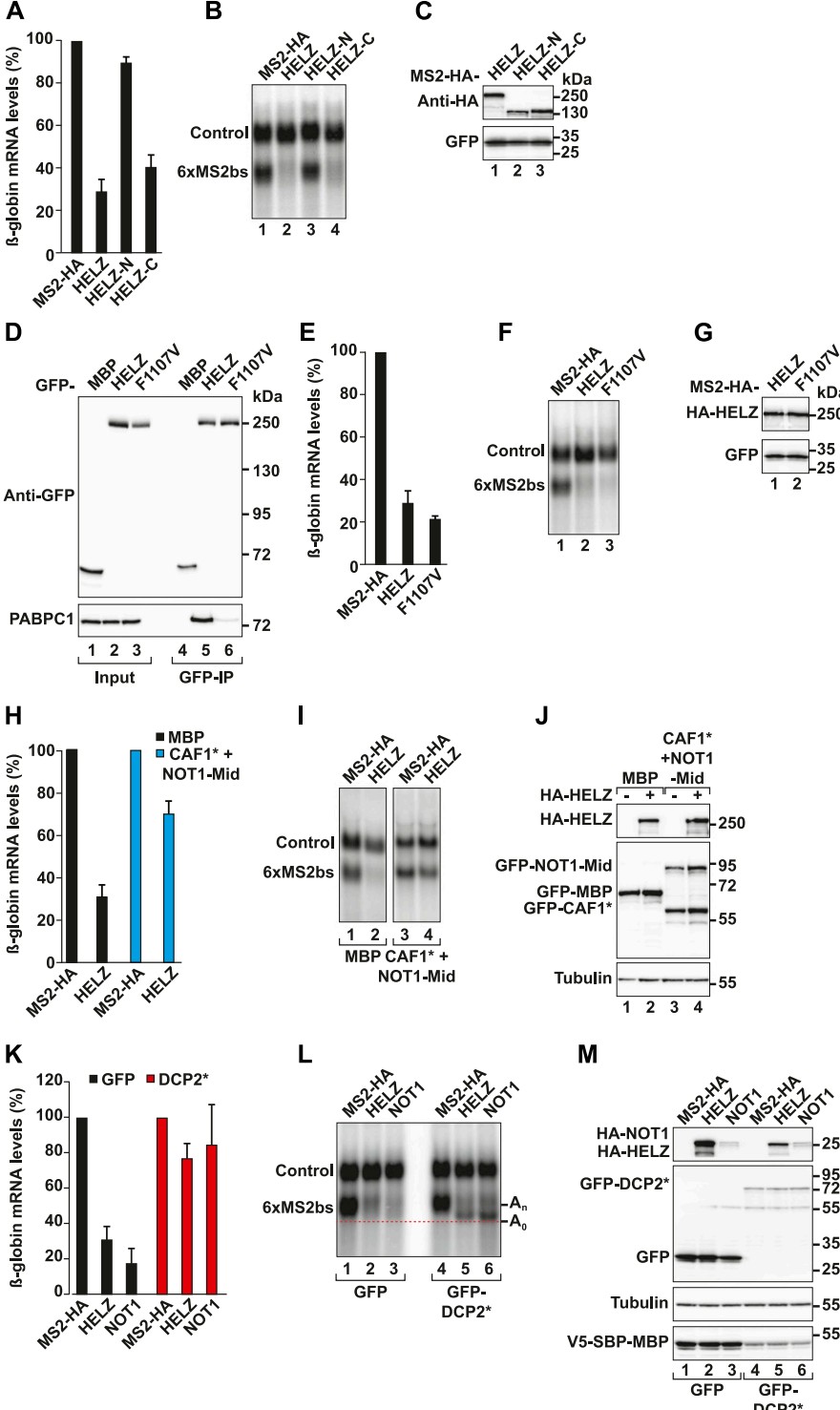

**Figure 3. HELZ induces 5′-to-3′ decay of tethered reporter mRNAs.**
**(A)** Tethering assay in HEK293T cells using the β-globin-6xMS2bs reporter and MS2-HA–tagged HELZ (full-length or indicated fragments). The control reporter lacking the MS2bs (control) contains the β-globin gene fused to a fragment of the *gapdh* gene. The graph shows the quantification of mRNA levels of the β-globin-6xMS2bs reporter normalized to the levels of the control reporter and set to 100 for MS2-HA; the mean values ± SD are shown for four independent experiments. **(B)** Representative Northern blot of samples shown in (A). **(C)** Representative Western blot depicting the equivalent expression of the MS2-HA–tagged proteins used in (A) and (B). GFP served as a transfection control. **(D)** Immunoprecipitation assay in HEK293T cells showing the interaction of GFP-tagged HELZ wild-type (WT) and F1107V mutant with endogenous PABPC1. GFP-MBP was used as a negative control. Input (1.2%) and bound fractions (20% for GFP-tagged proteins and 35% for endogenous PABPC1) were analysed by Western blotting. **(E)** Tethering assay as described in (A), in cells expressing MS2-HA–tagged HELZ WT and F1107V mutant as indicated. The mean values ± SD are shown for four independent experiments. **(F)** Representative Northern blot of samples used in (E). **(G)** Western blot depicting the equivalent expression of the MS2-HA-HELZ WT and F110V in (E) and (F). GFP served as a transfection control. **(H)** Tethering assay as described in (A), but the transfection mixture included additionally plasmids expressing GFP-CAF1* and GFP-NOT1-Mid to block deadenylation (blue bars). GFP-MBP was overexpressed in control samples (black bars). The mean values ± SD are shown for three independent experiments. **(I)** Northern blot with representative RNA samples from the experiment depicted in (H). **(J)** Western blot showing the equivalent expression of HA-HELZ and the GFP-tagged proteins used in (H) and (I). Tubulin served as loading control. **(K)** Tethering assay as described in (A). The transfection mixture additionally included a plasmid expressing GFP-DCP2* catalytic mutant to block decapping (red bars). GFP was overexpressed in control samples (black bars). Tethering of MS2-HA-NOT1 served as positive control for deadenylation-dependent decapping (Kuzuoglu-Ozturk et al, 2016). The mean values ± SD are shown for three independent experiments. **(L)** Northern blot of representative RNA samples corresponding to the experiment shown in (K). The position of the fast migrating deadenylated form of the reporter mRNA ($A_0$) is marked with a red dotted line, whereas the position of the reporter with an intact poly(A) is indicated as ($A_n$). **(M)** Western blot showing the expression of HA-HELZ, HA-NOT1, and the GFP-tagged proteins used in (K) and (L). Tubulin served as loading control and V5-SBP-MBP as a transfection control. Transfection efficiency and/or plasmid expression was decreased in cells expressing GFP-DCP2*.
Source data are available for this figure.

Omega/Align [Pundir et al, 2016]). Similar to the human orthologue, GFP-tagged *Dm* HELZ immunoprecipitated various mRNA decay factors when expressed in *Dm* Schneider S2 cells, including *Dm* HPat (fly orthologue of mammalian PatL1) and *Dm* PAN3 (Fig 4A and B), but not *Dm* Ge-1 (fly orthologue of mammalian EDC4) or *Dm* Me31B (fly orthologue of mammalian DDX6) (Fig S1B and C). GFP-tagged *Dm* HELZ

also immunoprecipitated the *Dm* CCR4–NOT complex proteins NOT1 and NOT2 (Fig 4C and D), indicating that these interactions are a conserved feature of HELZ orthologues.

Next, we tested whether *Dm* HELZ can induce mRNA decay. We used a λN-based tethering assay to recruit λN-HA-tagged *Dm* HELZ full-length protein or fragments to a firefly luciferase reporter

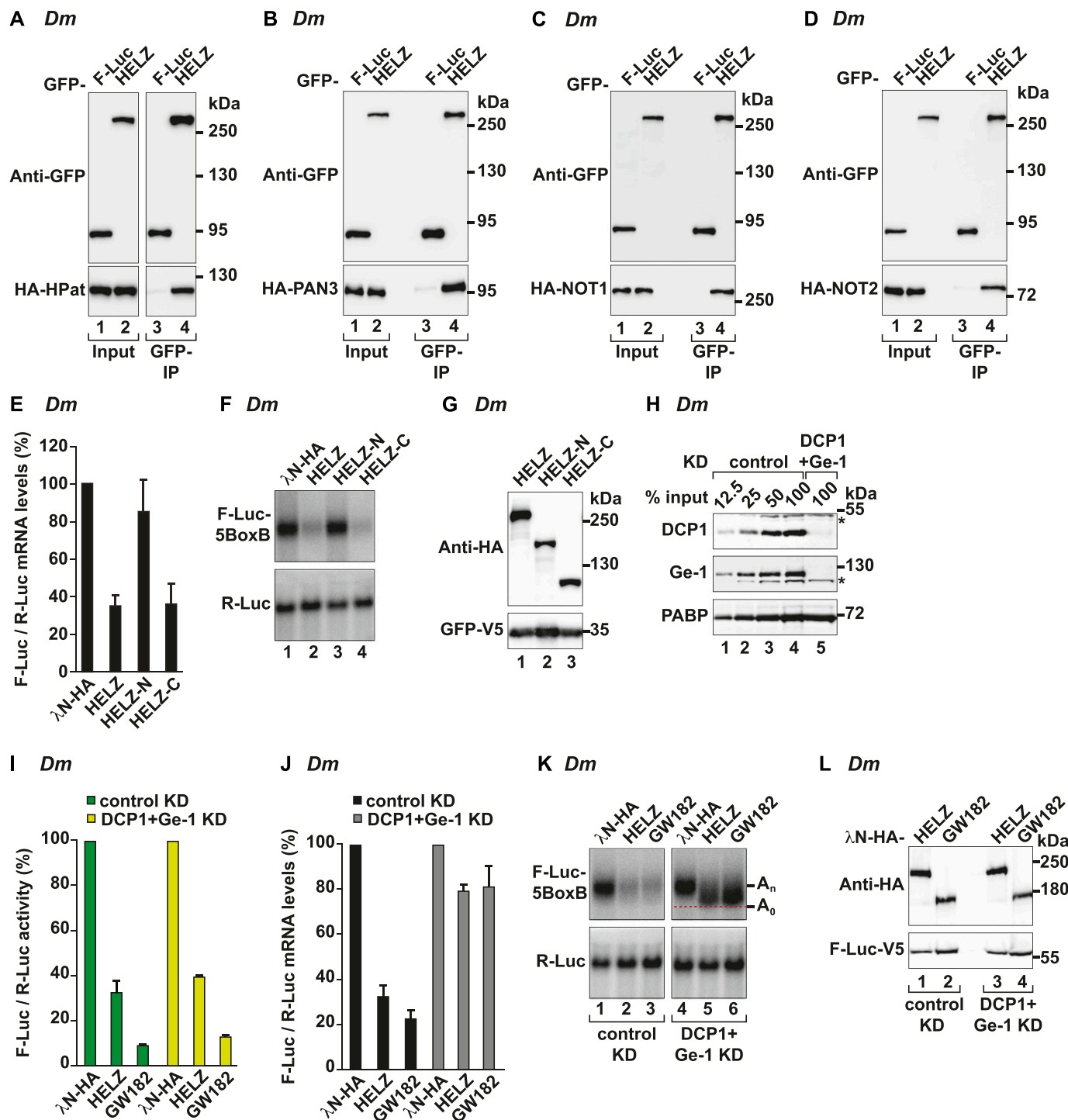

**Figure 4.   The role of HELZ in inducing mRNA decay is conserved in Metazoa.**
**(A–D)** Immunoprecipitation assays in *Dm* S2 cells showing the interaction of GFP-*Dm* HELZ with HA-tagged-*Dm* HPat (A), HA-tagged-*Dm* PAN3 (B), HA-tagged-*Dm* NOT1 (C), and HA-tagged-*Dm* NOT2 (D). F-Luc-GFP served as negative control. Input (3.5% for GFP-tagged proteins and 0.5% for HA-tagged proteins) and bound fractions (10% for GFP-tagged proteins and 35% for HA-tagged proteins) were analysed by Western blotting. **(E)** Tethering assay in *Dm* S2 cells using the F-Luc-5BoxB reporter and λN-HA-*Dm* HELZ (full-length and fragments). A plasmid expressing R-Luc served as transfection control. F-Luc mRNA levels were normalized to those of the R-Luc control and set to 100 in cells expressing λN-HA. Graph shows the mean values ± SD of four experiments. **(F)** Representative Northern blot of samples shown in (E). **(G)** Western blot showing the equivalent expression of the λN-HA–tagged proteins used in (E). GFP-V5 was used as transfection control. **(H)** *Dm* S2 cells were treated with dsRNA targeting glutathione S-transferase (control) or DCP1 and Ge-1 mRNAs. The efficacy of the KD was estimated by Western blot with antibodies specifically recognizing endogenous DCP1 and Ge-1 proteins. PABP served as a loading control. Dilutions of control cell lysates were loaded in lanes 1–4 to estimate the efficacy of the depletion. The asterisks (*) mark unspecific bands recognized by the respective antibody. **(I, J)** *Dm* S2 cells treated with dsRNA targeting either glutathione S-transferase (control, green bars) or DCP1 and Ge-1 mRNAs (yellow bars) were transfected as described in (E). Tethering of λN-HA-GW182 served as positive control for deadenylation-dependent decapping

harboring five λN-binding sites (F-Luc-5xBoxB) in the 3′ UTR (Gehring et al, 2005; Behm-Ansmant et al, 2006). A reporter mRNA encoding *Renilla* luciferase (R-Luc) served as a transfection control. Tethering of *Dm* HELZ caused strong repression of the firefly luciferase activity compared with the control λN-HA protein (Fig S1D). Reporter mRNA levels were reduced in a similar manner (Fig 4E and F), indicating that the observed decrease in F-Luc activity was a consequence of mRNA decay. Interestingly, similar to the human orthologue, the C-terminal region of *Dm* HELZ (Table S1) was sufficient to elicit decay of the bound reporter. The *Dm* HELZ N-terminal fragment (Table S1) did not detectably impact on the stability of the F-Luc mRNA (Fig 4E–G) and instead stimulated F-Luc activity upon tethering (Fig S1D). The cause behind this observation is currently unclear.

To examine if *Dm* HELZ also induces deadenylation-dependent mRNA decapping, we performed tethering assays in *Dm* S2 cells depleted of two decapping activators DCP1 and Ge-1 to efficiently inhibit 5′-cap removal (Fig 4H–L; Eulalio et al, 2007b). In the absence of these decapping factors, tethering of HELZ to F-Luc-5BoxB resulted in a marked stabilization of the deadenylated variant of the reporter transcript (Fig 4J and K, lane 5). Similar results were obtained with tethered GW182 (Fig 4J and K, lane 6), which triggers deadenylation-dependent decapping and thus served as a positive control (Behm-Ansmant et al, 2006). The inhibition of decapping and the resulting stabilization of the deadenylated reporter did not lead to the restoration of F-Luc protein levels consistent with impaired translation of the reporter mRNA lacking a poly(A) tail (Fig 4I). We conclude that in *Dm*, as in human cells, HELZ interacts with components of the mRNA decay machinery and promotes decapping-dependent decay of a bound mRNA.

### HELZ requires CCR4–NOT to repress translation of bound mRNAs

We then investigated if HELZ can repress translation in the absence of deadenylation. We used an R-Luc reporter mRNA that does not undergo deadenylation and subsequent decay (R-Luc-6xMS2bs-$A_{95}$-MALAT1) (Bhandari et al, 2014; Kuzuoglu-Ozturk et al, 2016). This reporter harbors a 95-nt internal poly(A) stretch followed by the 3′-terminal region of the metastasis associated lung adenocarcinoma transcript 1 (MALAT1) noncoding RNA, which is processed by RNaseP and thus lacks a poly(A) tail (Wilusz et al, 2012). An F-Luc-GFP reporter served as a transfection control. In the presence of HELZ, R-Luc activity was reduced to 40% relative to MS2-HA without changes in mRNA levels (Fig 5A–C, lane 2). This result indicates that deadenylation is not required for HELZ-mediated translational repression. Interestingly, the HELZ (F1107V) mutant, which cannot interact with PABPC1, was equally active to WT HELZ in eliciting deadenylation-independent translational repression (Fig 5A–D).

The CCR4–NOT complex not only mediates deadenylation but can also promote translational repression of target mRNAs (Cooke et al, 2010; Chekulaeva et al, 2011; Bawankar et al, 2013; Zekri et al, 2013). To address if HELZ-mediated translational repression depends

on the CCR4–NOT complex, we tethered HELZ to the R-Luc reporter in HeLa cells depleted of NOT1. shRNA-mediated knock-down (KD) resulted in a pronounced reduction of NOT1 protein levels without affecting MS2-HA-HELZ expression (Fig 5E, lanes 4 and 5). NOT1 depletion, however, severely compromised the ability of HELZ to repress the translation of the R-Luc-6xMS2bs-$A_{95}$-MALAT1 reporter (Fig 5F), consistent with the function of HELZ as a translational repressor being dependent on the CCR4–NOT complex.

Repression of translation by the CCR4–NOT complex is strongly associated with the DEAD-box helicase DDX6, a decapping activator and an inhibitor of translation (Maillet & Collart, 2002; Chu & Rana, 2006; Chen et al, 2014; Mathys et al, 2014; Freimer et al, 2018). To probe for this molecular connection in the context of translational repression by HELZ, we generated a HEK293T *Ddx6*-null cell line using CRISPR-Cas9 genome editing. Successful gene targeting was verified by the loss of DDX6 protein expression and genomic DNA sequencing of the targeted exon (Fig S2A and see the Materials and Methods section). Characterization of the *Ddx6*-null cells by polysome profiling indicated that DDX6 depletion does not induce major changes in general translation in HEK293T cells cultured under standard conditions (Fig S2B) relative to wild type (WT) cells. DDX6 depletion did, however, result in a drastic reduction of P-bodies as shown by the abnormal distribution of the P-body component EDC4 (Fig S2C and D) and as previously reported (Lumb et al, 2017; Freimer et al, 2018).

In the absence of DDX6, translational repression of the R-Luc-6xMS2bs-$A_{95}$-MALAT1 reporter by HELZ was impaired, albeit not completely abolished, as R-Luc activity recovered from 45% in WT cells to 70% in the *Ddx6*-null cells (Fig 5G). In contrast, loss of DDX6 did not change the ability of the silencing domain of TNRC6A (TNRC6A-SD; Lazzaretti et al, 2009) to repress the expression of the MALAT1 reporter (Fig 5E and F). Furthermore, exogenous expression of GFP-DDX6 restored HELZ repressive activity in *Ddx6*-null cells (Fig 5G and H). Comparable MS2-HA-HELZ protein levels in WT and DDX6-complemented cells were confirmed by Western blotting (Fig 5H). Thus, DDX6 is involved in HELZ-mediated translational repression.

### HELZ is not required for CCR4–NOT–mediated translational repression and mRNA decay

To further address the role of HELZ in mRNA metabolism, we generated a *Helz*-null HEK293T cell line using CRISPR-Cas9 gene editing (Fig S2G). *Helz*-null cells proliferated at normal rates, and no changes were observed in general translation, as assessed by polysome profiling analysis (Fig S2H). Furthermore, in these cells, the protein levels of the CCR4–NOT components NOT2 and NOT3, PABPC1, as well as DDX6 were similar to WT cells (Fig S2G).

We then tested if NOT1-mediated posttranscriptional gene regulation is impaired in the absence of an interaction with HELZ.

(Behm-Ansmant et al, 2006). Panel (I) shows relative F-Luc activity in control and DCP1 + Ge-1 KD samples. Panel (J) depicts relative F-Luc mRNA levels in control and DCP1 + Ge-1 KD samples. The mean values ± SD are shown for five independent experiments. **(K)** Representative Northern blot analysis of samples shown in (J). The position of the fast migrating deadenylated form of the reporter mRNA ($A_0$) is marked with a red dotted line, whereas the position of the reporter mRNA with intact poly(A) is indicated as ($A_n$). **(L)** Western blot showing the equivalent expression of the λN-HA–tagged proteins used in (I). F-Luc-V5 was used as transfection control. Source data are available for this figure.

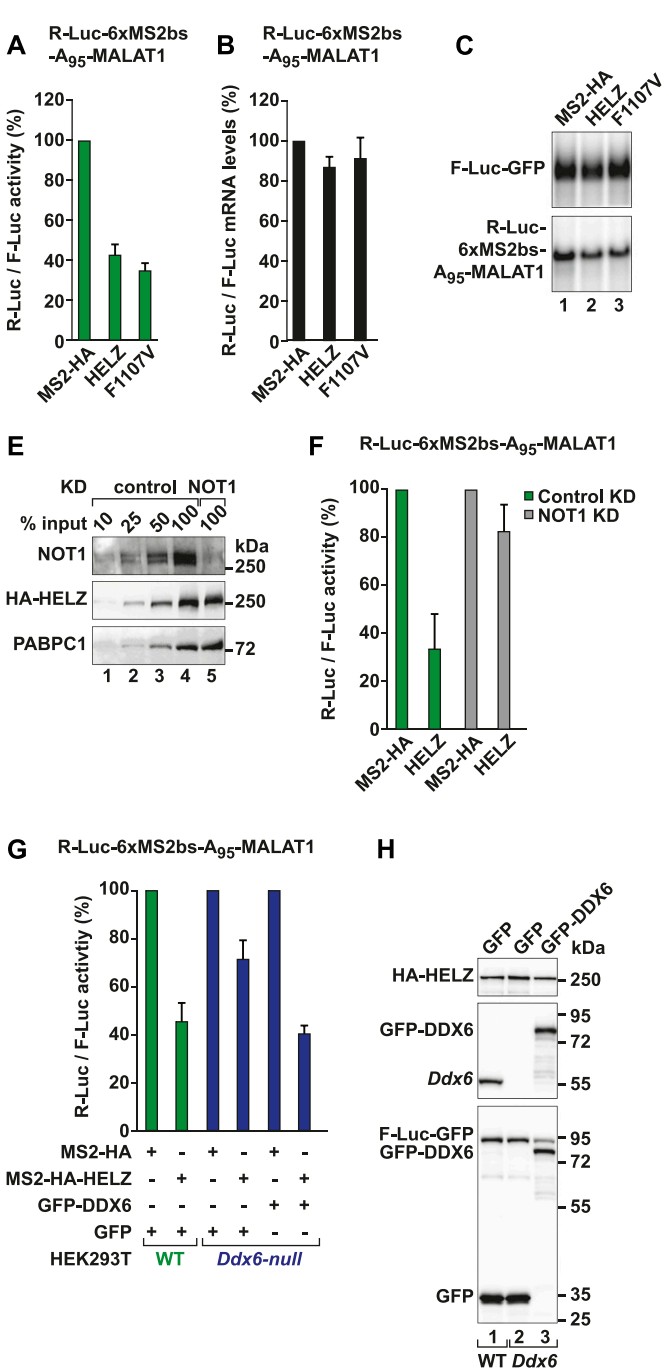

**Figure 5. HELZ requires CCR4–NOT to repress translation of bound mRNAs.**

**(A, B)** Tethering assay in HEK293T cells using the R-Luc-6xMS2bs-$A_{95}$-MALAT1 reporter with MS2-HA-HELZ WT and F1107V mutant. A plasmid coding for F-Luc-GFP served as control. Shown is the quantification of protein (A) and of mRNA levels (B) of the R-Luc-6xMS2bs-$A_{95}$-MALAT1 reporter normalized to the levels of the control reporter and set to 100 for MS2-HA. The mean values ± SD are shown for four independent experiments. **(C)** Representative Northern blots of samples shown in (B). **(D)** Western blot showing the equivalent expression of the MS2-HA tagged proteins used in (A). F-Luc-GFP was used as transfection control. **(E)** Western blot analysis of HeLa cells after NOT1 KD. Dilutions of control cell lysates were loaded in lanes 1–4 to estimate the efficacy of NOT1 depletion. Transfected MS2-HA-HELZ protein was expressed at comparable levels in WT and NOT1 KD cells. PABPC1 served as a loading control. **(F)** Tethering assay in HeLa cells using the R-Luc-6xMS2bs-$A_{95}$-MALAT1 reporter and MS2-HA-HELZ. HeLa cells were treated with scrambled shRNA (green bar) or shRNA targeting NOT1 mRNA (grey bar). The graph shows relative R-Luc activity in control and NOT1 KD samples. The mean values ± SD are shown for three independent experiments. **(G)** Tethering assay in HEK293T WT (green bars) and *Ddx6*-null cells (blue bars) with MS2-HA-HELZ and the R-Luc-6xMS2bs-$A_{95}$-MALAT1 reporter. For complementation studies, the cells were also transfected with either GFP or GFP-DDX6. A plasmid expressing F-Luc-GFP served as a transfection control. Shown is the quantification of R-Luc activity normalized to F-Luc activity and set to 100 for MS2-HA in WT or *Ddx6*-null cells. The mean values ± SD are shown for three independent experiments. **(H)** Western blot showing the levels of transfected MS2-HA-HELZ protein in the different cell lines used in (G). Loss of endogenous DDX6 protein expression in HEK293T *Ddx6*-null cells was confirmed using an anti-DDX6 antibody (lane 2, middle panel). The blot further illustrates that GFP-DDX6 was expressed at a level equivalent to endogenous DDX6 (lane 3 versus lane 1). F-Luc-GFP served as transfection control.

Source data are available for this figure.

Therefore, we tethered NOT1 to the R-Luc-6xMS2bs or the R-Luc-6xMS2bs-$A_{95}$-MALAT1 reporters in *Helz*-null cells. These reporters are degraded or translationally repressed, respectively, when bound to NOT1 (Kuzuoglu-Ozturk et al, 2016). Tethered NOT1 reduced R-Luc activity of both mRNA reporters to 20% in WT and *Helz*-null cells (Fig S3A–E). These results are in agreement with HELZ acting upstream of the deadenylase complex (i.e., as a recruitment factor). The more likely scenario is that HELZ acts together with the CCR4–NOT to regulate the expression of a subset of mRNAs.

## HELZ regulates the abundance of mRNAs encoding proteins involved in neurogenesis and nervous system development

To gain more insight into HELZ mRNA targets, we next investigated how the cellular transcriptome is affected in the absence of HELZ. Thus, we sequenced and analysed the transcriptome of the *Helz*-null and WT cells (Figs 6A and S4 and Table S2). The replicates of the RNA-Seq libraries of the two cell types clustered together as determined using multidimensional scaling analysis (Fig S4A). HELZ depletion induced major changes in the cellular transcriptome.

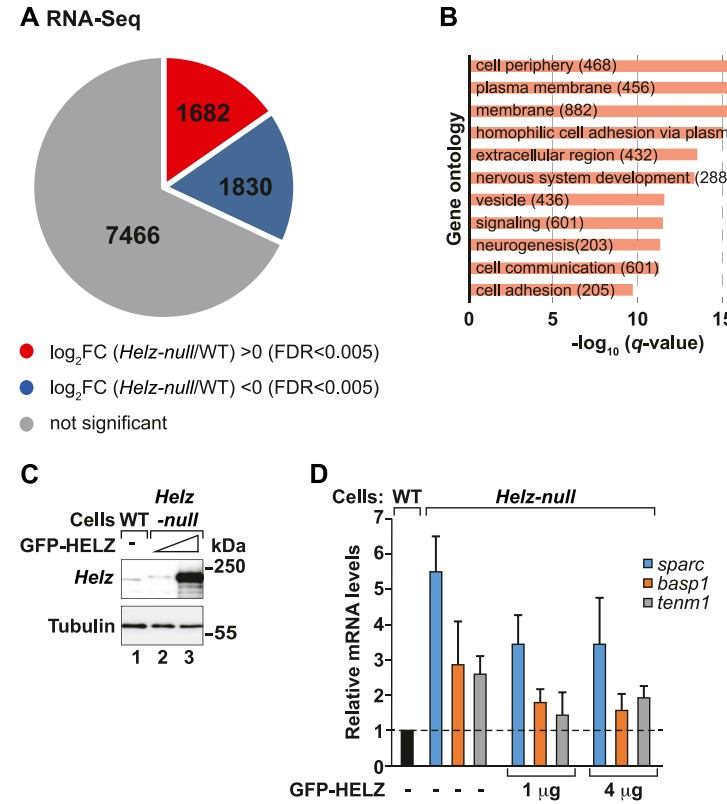

**A** RNA-Seq

- 🔴 log$_2$FC (*Helz-null*/WT) >0 (FDR<0.005)
- 🔵 log$_2$FC (*Helz-null*/WT) <0 (FDR<0.005)
- ⚪ not significant

**Figure 6. Transcriptome analysis of HEK293T *Helz*-null cells.**
**(A)** Pie chart indicating the fractions and absolute numbers of differentially expressed genes derived from the analysis of the transcriptome of HEK293T wild-type (WT) and *Helz*-null cells by RNA-Seq. Two biological replicates of each cell line were analysed. The RNA-Seq analysis indicated that 7,466 (grey) of the total 10,978 genes selected using fragments per kilobase of transcript per million mapped reads >2 cut-off showed no significant differences between the two cell lines (FDR ≥ 0.005). 1,682 genes were significantly up-regulated (red) whilst 1,830 genes were down-regulated (blue) using an fold change (FC) >0 on log$_2$ scale with an FDR < 0.005 to determine abundance. **(B)** Gene ontology analysis of the biological processes overrepresented in the group of transcripts up-regulated in *Helz*-null cells (log$_2$FC > 0, FDR < 0.005) versus all other expressed genes. Bar graph shows –log$_{10}$ of q values for each category. Content of brackets indicates the number of genes within each category. **(C)** Western blot analysis depicting the levels of endogenous HELZ present in HEK293T WT cells (lane 1) compared with *Helz*-null cells transfected with either 1 or 4 µg of GFP-HELZ (lanes 2 and 3, respectively). Tubulin served as loading control. **(D)** qPCR validation of three up-regulated (log$_2$FC > 0, FDR < 0.005) transcripts identified in (A). Transcript levels of *sparc* (blue bars), *basp1* (orange bars) and *tenm1* (grey bars) were determined in HEK293T WT, *Helz*-null, and *Helz*-null cells complemented with either 1 or 4 µg of GFP-HELZ. Transcript levels were normalized to *gapdh* mRNA. Shown are the normalized expression ratios ± SD for three independent experiments.

In fact, differential gene expression analysis revealed 1,682 mRNAs to be significantly up-regulated (log$_2$FC > 0 and FDR < 0.005) and 1,830 mRNAs to be down-regulated (log$_2$FC < 0 and FDR < 0.005) in the *Helz*-null cells relative to WT cells (Figs 6A and S4B).

Functional annotation analysis using the goseq R-package (Young et al, 2010) for all up-regulated transcripts in *Helz*-null cells indicated a significant enrichment for genes encoding cell periphery (22%, q < 9.19 × 10$^{-20}$), membrane-associated (17%, q < 1.19 × 10$^{-16}$), cell adhesion (23%, q < 2.05 × 10$^{-10}$), and signalling (19%, q < 3.01 × 10$^{-12}$)-related proteins. Interestingly, many of the corresponding proteins have known functions in the biological processes of neurogenesis (25%, q < 5.06 × 10$^{-12}$) and nervous system development (23%, q < 4.5 × 10$^{-14}$). These include, for instance, GDNF (glial cell-line–derived neurotrophic factor) family receptor alpha-3 (GFRA3; Baloh et al, 1998; Naveilhan et al, 1998), brain acid soluble protein 1 (BASP1; Hartl & Schneider, 2019), teneurin (TENM1; Tucker, 2018), neurofilament medium polypeptide (NEFM; Coulombe et al, 2001) or the protocadherin G cluster (PCDHG; Keeler et al, 2015), among others (Table S3). After analysis of transcript length and nucleotide composition, we also observed that the mRNAs with increased abundance in the absence of HELZ have longer coding sequences (CDS; P < 2.2 × 10$^{-16}$) and a higher guanine and cytosine (GC) content across the whole gene (P = 6.1 × 10$^{-11}$ or P < 2.2 × 10$^{-16}$) compared to all other genes expressed in these cells (down-regulated mRNAs and all mRNAs not significantly altered in *Helz*-null cells, Fig S5).

On the other hand, transcripts with decreased expression in *Helz*-null cells were related to translation (structural constituent of ribosome [45%, q < 2.64 × 10$^{-21}$], signal-recognition particle-dependent cotranslational protein targeting to membrane [59%, q < 1.7 × 10$^{-20}$], ribosome biogenesis [36%, q < 2.99 × 10$^{-17}$], translation [43%, q < 3.66 × 10$^{-16}$], and rRNA processing [37%, q < 7.08 × 10$^{-16}$]). Other down-regulated and overrepresented GO terms included RNA metabolism and RNA-binding (RNP complex biogenesis [31%, q < 1.08 × 10$^{-15}$], nonsense-mediated decay [47%, q < 9.19 × 10$^{-20}$], non-coding RNA (ncRNA)-metabolic process [28%, q < 6.96 × 10$^{-13}$], and RNA binding [23%, q < 2.97 × 10$^{-11}$]) or organonitrogen compound metabolism (24%, q < 3.15 × 10$^{-16}$; Fig S4C).

To validate that the differentially expressed mRNAs identified in this analysis are indeed regulated by HELZ, we measured the abundance of three significantly up-regulated (FC > twofold, FDR < 0.005) transcripts in *Helz*-null cells upon transient expression of increasing concentrations of GFP-tagged HELZ (Fig 6C). In *Helz*-null cells, *sparc*, *basp1*, and *tenm1* mRNA levels, determined by quantitative RT-PCR (RT-qPCR), were increased relative to WT cells (Fig 6D), as observed in the RNA-Seq analysis (Table S2). Transcript levels increased 2.5–5.5 fold, depending on the mRNA. Overexpression of GFP-HELZ decreased the abundance of these transcripts, partially restoring steady state mRNA levels (Fig 6D).

These results suggest that HELZ has an important role in the control of the expression of specific genes. Increased transcript abundance can be explained by the activity of HELZ as a transcriptional (Hamamoto et al, 2004) and/or posttranscriptional regulator via its interaction with the CCR4–NOT complex (this study and [Mathys et al, 2014]). Additional studies are required to identify the transcripts co-regulated by HELZ and the CCR4–NOT complex.

## Discussion

The putative SF1 helicase HELZ has been associated with various steps in RNA metabolism, including transcription and translation. Here, we reveal that HELZ also regulates mRNA stability as it induces deadenylation and decapping of bound reporter mRNAs. This function is likely the result of HELZ interaction with various mRNA decay factors including components of the CCR4–NOT complex in human and *Drosophila* cells. In fact, human HELZ has multiple binding sites within its nonconserved and unstructured C-terminal region that directly interact with the NOT module of the CCR4–NOT complex (Fig 2). The NOT module, composed of NOT1/2/3 subunits is a known binding platform for various mRNA-associated proteins, including the posttranscriptional RNA regulator Nanos (Bhandari et al, 2014; Raisch et al, 2016) and the transcription factor E26-related gene (Rambout et al, 2016). Tethering of *Hs* and *Dm* HELZ to an mRNA reporter triggers decapping-dependent mRNA decay. In both species, the C-terminal region of HELZ was necessary and sufficient to elicit decay. The observation that the regulatory effect of HELZ on stability and translation of tethered mRNA requires the CCR4–NOT complex (Figs 3H–J and 5F) supports the functional connection between HELZ and CCR4–NOT in mRNA metabolism.

Recruitment of the CCR4–NOT complex to mRNA targets by short linear motifs (SLiMs) located in unstructured and poorly conserved regions of RNA-associated proteins is a common and widespread mechanism (Fabian et al, 2013; Bhandari et al, 2014; Raisch et al, 2016; Sgromo et al, 2017; Keskeny et al, 2019). The presence of multiple binding sites in the HELZ C-terminal region indicates a SLiM-mediated mode for interaction with the CCR4–NOT complex. The plastic evolutionary nature of SLiM-mediated protein binding (Davey et al, 2012; Tompa, 2012) readily explains how largely divergent and unstructured C-terminal regions of HELZ orthologues perform equivalent cellular functions.

Interestingly, HELZ is not the only SF1 helicase known to interact with the CCR4–NOT complex and promote mRNA decay. The UPF1 RNA helicase, through both direct and indirect interactions, binds to different mRNA decay factors, including the endoribonuclease SMG6 and the CAF1 deadenylase to induce target mRNA decay (Kim & Maquat, 2019). UPF1 contains a helicase core domain that is structurally highly similar to HELZ. UPF1 binds rather nonspecifically to accessible mRNAs (Zund & Muhlemann, 2013) but seems to be recruited through interaction with specific RNA-binding proteins to defined targets to participate in distinct mRNA decay pathways (Kim & Maquat, 2019). Whether HELZ function is subject to similar control is unknown.

Our study also highlights a potential role for HELZ as a translational repressor (Fig 5). HELZ-mediated translational repression of a reporter mRNA lacking a 3' poly(A) tail depends on the CCR4–NOT complex but does not require binding to PABPC1. Repression of translation by the CCR4–NOT complex is associated with the DEAD-box helicase DDX6 (Maillet & Collart, 2002; Chu & Rana, 2006; Chen et al, 2014; Mathys et al, 2014; Freimer et al, 2018), and we provide evidence that DDX6 contributes to HELZ-induced translational repression. However, in the absence of DDX6, the translational repressor function of HELZ was not completely abolished. Thus, other factors are involved in HELZ-mediated translational

repression. Another HELZ- and CCR4–NOT–interacting protein is the translational repressor PatL1 (Fig 1C) (Braun et al, 2010; Ozgur et al, 2010) and additional studies will determine the relevance of PatL1, or other factors, in the repression of translation by HELZ and the CCR4–NOT complex.

HELZ contains several sequence motifs that could confer RNA binding ability. Its PABPC1 binding property suggests that HELZ has a preference for polyadenylated mRNAs. Furthermore, HELZ contains a CCCH-type ZnF motif in the N terminus (Fig 1A) that may be critical for its biological role as it can promote protein–protein interactions or facilitate RNA recognition (Hall, 2005; Gamsjaeger et al, 2007). This specific type of ZnF is present in RNA-binding proteins such as tristetraprolin and Roquin, which also directly recruit the CCR4–NOT complex to mRNA targets, promoting their degradation (Fabian et al, 2013; Fu & Blackshear, 2017; Sgromo et al, 2017).

Although it remains unclear how HELZ is recruited to mRNA, transcriptome-wide analysis of *Helz*-null cells via RNA-Seq indicated that HELZ depletion has a substantial impact on gene expression (Figs 6 and S4). Interestingly, genes with up-regulated expression in the absence of HELZ code for membrane- and cell periphery–associated proteins, many of which participate in the development of the nervous system (Fig 6B and Table S3). An important goal for future studies is to investigate HELZ and its association with the CCR4–NOT complex in the posttranscriptional regulation of this biological process.

HELZ loss also resulted in decreased abundance of transcripts with gene products involved in translation. Even if global translation was not altered in *Helz*-null cells (Fig S2H), this observation is in line with the fact that HELZ overexpression results in increased translation and cellular proliferation (Hasgall et al, 2011). Moreover, similar to HELZ depletion, loss of the HELZ-interacting protein and transcriptional regulator Smyd2 in cardiomyocytes leads to decreased expression of genes functionally associated with translation (Diehl et al, 2010).

In conclusion, our findings support a role of HELZ as a regulator of gene expression and highlight a potential development- or cell-specific function for this RNA helicase. Furthermore, the direct interaction of HELZ with the CCR4–NOT complex described in this study represents another molecular mechanism used by HELZ in the control of gene expression.

## Materials and Methods

### DNA constructs

All the mutants used in this study were generated by site-directed mutagenesis using the QuickChange mutagenesis kit (Stratagene). All the constructs and mutations were confirmed by sequencing and are listed in Table S1. To generate the pT7-EGFP-*Hs* CAF1* catalytic mutant, D40A and E42A point mutations were introduced into the pT7-EGFP-*Hs* CAF1 vector (Braun et al, 2011). *Hs* HELZ cDNA was amplified from the Kazusa clone KIAA0054 and inserted into the SacII and SalI restriction sites of the pT7-EGFP-C1 vector or the SacII and XbaI restriction sites of the pT7-MS2-HA vector. For MS2-HA–tagged *Hs* HELZ proteins, the pT7-λN-HA-C1 vector was modified by

mutagenesis to replace the λN-HA-tag with the MS2-HA-tag. The *Hs* HELZ-N and HELZ-C fragments (residues M1–D1050 and P1051–K1942, respectively) were amplified by PCR using specific primers (*Hs* HELZ-N: forward: ATACATCCGCGGATATGGAAGACAGAAGAGCTGAAAAGT, reverse: ACATTCTAGATTAATCACCCACCACAGCAACCAGGGAT; *Hs* HELZ-C: forward: ATACATCCGCGGATCCCATTGCTCTGTGCTCTATTGGAA, reverse: ACATTCTA-GATTATTTAAAATATGAGTAAAAGCCA) and inserted between the restrictions sites SacII and XbaI of the pT7-EGFP-C1 and pT7-MS2-HA-C1 vectors. The *Hs* NOT1 ORF was amplified from cDNA and inserted into the XhoI and SacII sites of the pT7-MS2-HA vector. The plasmid allowing the expression of HA-*Hs* DDX6 was generated by cloning the corresponding cDNA into the XhoI and NotI restriction sites of the pCIneo-λN-HA vector. To obtain the plasmid expressing the silencing domain of *Hs* TNRC6A (residues T1210–V1709), the corresponding cDNA amplified by PCR was cloned into the BamHI and XhoI restriction sites of the pcDNA3.1-MS2-HA vector. The plasmids for the expression of the HA-tagged versions of *Hs* EDC4, *Hs* PatL1, and *Hs* PAN3 or *Dm* HPat, *Dm* PAN3, *Dm* NOT1, *Dm* NOT2, *Dm* Ge-1, and *Dm* Me31B were previously described (Eulalio et al, 2007a; Tritschler et al, 2008, 2009; Braun et al, 2010, 2011; Bawankar et al, 2013).

*Dm* HELZ was amplified from cDNA derived from S2 cells and inserted into the pAc5.1B-λN-HA and pAc5.1B-EGFP vectors between HindIII and XbaI restriction sites (Eulalio et al, 2007a). *Dm* HELZ-N and HELZ-C (residues M1–D1212 and P1213–Q2103, respectively) were amplified by PCR using specific primers (*Dm* HELZ-N: forward: ATACATAAGCTTCATGGCCGCCGAGAAGGAGATGCAGGC, reverse: ACATTC-TAGATTAATCACCAACCACTGCAACCAACGAC; *Dm* HELZ-C: forward: ATACATAAGCTTCCCCGTGGCTCTTTGTTCCATTGGTC, reverse: ACATTC-TAGATTACTGAAAATAGTTGTAGAATCCG) and inserted between the restriction sites HindIII and XbaI of the pAc5.1B-λN-HA plasmid.

For expression of recombinant *Hs* HELZ-C1 and HELZ-C2 in bacteria, the corresponding sequences were amplified by PCR and inserted between the BspTI and XbaI restriction sites of the pnEA-NvM plasmid (Diebold et al, 2011), resulting in HELZ fusion proteins carrying an N-terminal MBP tag cleavable by the tobacco etch virus protease. In addition, the DNA encoding the B1 domain of immunoglobulin-binding protein G (GB1) (Cheng & Patel, 2004), followed by a four-residue long (Met-Gly-Ser-Ser) linker sequence and a hexa histine (His$_6$)-tag were added to the end of the HELZ-C1 and HELZ-C2 coding sequences by site-directed mutagenesis.

## Tethering assays

The reporter constructs used in the tethering assays performed in human and *Dm* cells were described previously (Lykke-Andersen et al, 2000; Behm-Ansmant et al, 2006; Kuzuoglu-Ozturk et al, 2016). In the case of tethering assays in HEK293T WT, *Helz*-null and *Ddx6*-null, and HeLa cell lines, cells were cultured in 6-well plates and transfected using Lipofectamine 2000 (Invitrogen) according to the manufacturer's recommendation. The transfection mixture used in Fig 3A and E contained the following plasmids: 0.5 µg control β-globin, 0.5 µg β-globin-6xMS2bs and the following amounts of the plasmids expressing the MS2-HA–tagged proteins: 1 µg of *Hs* HELZ and *Hs* HELZ F1107V, 1.35 µg of *Hs* HELZ-N, and 2.5 µg of *Hs* HELZ-C. In Fig 3H, the transfection mixtures contained, in addition, plasmids expressing GFP-MBP (2 µg) or GFP-*Hs* CAF1* (1 µg) together with GFP-*Hs* NOT1-Mid region (residues M1085–T1605; 1 µg) (Petit et al, 2012).

In Fig 3K, the transfection mixtures contained, in addition, plasmids expressing GFP (0.15 µg) or GFP-*Hs* DCP2* (2 µg) (Chang et al, 2014). In the tethering assays with luciferase (R-Luc and F-Luc) reporters depicted in Figs 5 and S3, the transfection mixtures contained 0.2 µg F-Luc-GFP (transfection control), 0.2 µg of R-Luc-6xMS2bs (or R-Luc), or 0.5 µg R-Luc-6xMS2bs-A$_{95}$-MALAT1 (or R-Luc-A$_{95}$-MALAT1) and 1 µg of MS2-HA-*Hs* HELZ or MS2-HA-*Hs* NOT1. The transfection mixture in the experiment described in Fig 5G additionally contained 0.2 µg of the plasmid required for the expression of GFP-DDX6 in *Ddx6*-null cells. The cells were harvested 2 d after transfection for further analysis. shRNA-mediated KD of NOT1 in HeLa cells was performed as previously described (Chen et al, 2014). In the experiment described in Fig S2E, the transfection mixture contained 0.5 µg of MS2-HA or MS2-HA-TNRC6A-SD, 0.5 µg of R-Luc-6xMS2bs-MALAT1, and 0.5 µg F-Luc-GFP (transfection control).

To perform tethering assays with *Dm* HELZ, S2 cells were seeded in 6-well plates and transfected with Effectene Transfection Reagent (QIAGEN) according to the manufacturer's recommendation. The transfection mixture contained 0.4 µg of R-Luc, 0.1 µg of F-Luc-V5, or F-Luc-5BoxB and 0.01 µg of λN-HA-GW182 or the following amounts of pAc5.1-λN-HA plasmids expressing *Dm* HELZ proteins: 0.4 µg HELZ, 0.2 µg HELZ-N, and 0.2 µg HELZ-C. RNAi-mediated KD of DCP1 and Ge-1 in *Dm* S2 cells was performed as described previously (Clemens et al, 2000; Zekri et al, 2013).

Total RNA was isolated using TriFast (Peqlab) and analysed by Northern blot as described previously (Behm-Ansmant et al, 2006). *Renilla* and firefly luciferase activities were measured using the Dual Luciferase Reporter Assay System (Promega).

## RNase H digestion

For the experiment depicted in Fig S1A, 10 µg of RNA was incubated with 3 µl of RNase H 5 U/µl (New England BioLabs) and 6 µM of oligo(dT) 15-mer in 100 µl H$_2$O for 1 h at 37°C and subsequently purified by phenol–chloroform extraction. The RNase H–treated RNA was then analysed via Northern blotting.

## Co-IP assays and Western blotting

Co-IP assays in human and *Dm* S2 cells were performed as previously described (Kuzuoglu-Ozturk et al, 2016). Briefly, for the human GFP-IP assays, 4 × 10$^6$ HEK293T cells were grown in 10-cm dishes and transfected the day after seeding using TurboFect transfection reagent (Thermo Fisher Scientific). The transfection mixtures in Fig 1B–E contained 15 µg of GFP-*Hs* HELZ and 10 µg of HA-EDC4, HA-PatL1, HA-PAN3, or HA-DDX6. The transfection mixtures in Fig 1F contained 20, 30, or 25 µg of plasmids expressing GFP-tagged *Hs* HELZ, *Hs* HELZ-N, or *Hs* HELZ-C, respectively.

The co-IP assays in S2 cells required two wells of a six-well plate (seeded at 2.5 × 10$^6$ cells per well) per condition. The cells were harvested 3 d after transfection with Effectene Transfection Reagent (QIAGEN). The transfection mixture contained 1 µg of GFP-*Dm* HELZ and 0.5 µg of HA-*Dm* Me31B, 1 µg of HA-*Dm* HPat, HA-*Dm* PAN3, HA-*Dm* NOT2, HA-*Dm* Ge-1, or 2 µg of HA-*Dm* NOT1.

All lysates were treated with RNase A before IP. Western blots were developed with the ECL Western Blotting Detection System

(GE Healthcare) according to the manufacturer's recommendations. Antibodies used in this study are listed in Table S4.

## Protein expression and purification

The purification of the human pentameric CCR4–NOT complex (CAF1/CAF40/NOT1/2/3) and the different modules was previously described (Sgromo et al, 2017). The pentameric CCR4–NOT complex comprises NOT1 (residues E1093–E2371), CAF1, CAF40 (residues R19–E285), NOT2 (residues T344–F540), and His$_6$-NOT3 (residues G607–Q753); the CAF1 module comprises NOT1 (residues E1093–S1317) and CAF1; the CAF40 module consists of NOT1 (residues V1351–L1588) and CAF40 (residues R19–E285); the MIF4G-C domain represents NOT1 residues Q1607–S1815; and the NOT module contains NOT1 (residues H1833–M2361), NOT2 (residues M350–F540), and NOT3 (residues L607–E748). *Hs* HELZ-C1 and *Hs* HELZ-C2 recombinant proteins were expressed with an N-terminal MBP- and a C-terminal GB1-His$_6$-tag in *Escherichia coli* BL21 (DE3) Star cells (Invitrogen) in Lysogeny broth (Luria broth) medium overnight at 20°C. The cells were sonicated in binding buffer containing 50 mM Hepes, pH 7, 200 mM NaCl, 20 mM imidazole, and 2 mM β-mercaptoethanol, supplemented with protease inhibitors, 1 mg/ml lysozyme, and 5 mg/ml DNase I. The cleared lysates were bound to an Ni$^{2+}$ HiTrap IMAC HP (GE Healthcare) column and proteins were eluted by a step gradient to binding buffer supplemented with 500 mM imidazole using Äkta Pure (GE Healthcare). The fractions in the single peak were analysed on an SDS–PAGE, pooled, and used in MBP pull-downs.

## In vitro MBP pull-down assays

Purified MBP (7.5 μg), MBP-*Hs* HELZ-C1-GB1-His or MBP-*Hs* HELZ-C2-GB1-His (500 μg each) were mixed with equimolar amounts of the purified CCR4–NOT subcomplexes in 1 ml of pull-down buffer (50 mM Hepes, pH 7, 200 mM NaCl, and 2 mM DTT) and incubated for 1 h at 4°C. After another hour of incubation at 4°C with 50 μl of amylose resin slurry (New England BioLabs), the beads were washed five times with pull-down buffer. The proteins were eluted with pull-down buffer supplemented with 25 mM D-(+)-maltose. The eluate was mixed 1:1 with 20% cold trichloroacetic acid (Roth) and incubated for 30 min on ice. The mix was then centrifuged at full speed at 4°C in a table-top centrifuge and the pellet was suspended in 35 μl of protein sample buffer (50 mM Tris–HCl, pH 6.8, 2% [wt/vol] SDS, 10% [vol/vol] glycerol, and 100 mM DTT). The eluted proteins were heated at 95°C for 5 min and analysed by SDS–PAGE. The gels were stained with Coomassie Blue overnight at room temperature and washed the next day.

## Generation of the HEK293T *Helz*- and *Ddx6*-null cell lines

The generation of the HEK293T HELZ- and *Ddx6*-null cell lines was essentially performed as described previously Sgromo et al, 2018. In the case of *ddx6*, a guide RNA targeting exon 2 (5′-GTCTTTTTCCAGTCATCACC-3′) was designed using DNA 2.0 (ATUM, www.atum.bio) online tool to minimize off-target effects. Genome targeting resulted in a 1-nt insertion in one allele and a 10-nt deletion in the other allele, both causing a frameshift of the ORF. To edit *helz* gene, a guide RNA targeting exon 8 (5′-GCAACTAGTAACGCCCTCTC-3′) was used.

*helz* gene targeting produced a 7-nt deletion causing a frameshift of the ORF.

## Transcriptome sequencing (RNA-Seq) and RT-qPCR validation

Total RNA was extracted from HEK293T WT or *Helz*-null cells using the RNeasy Mini Kit (QIAGEN) and a library prepared using the TruSeq RNA Sample Prep Kit (Illumina). Two biological replicates were analysed. RNA-Seq libraries were sequenced with the HiSeq 3000 sequencing system (Illumina) using paired-end sequencing. During data analysis, ribosomal RNA sequencing reads were filtered using Bowtie2 (Langmead & Salzberg, 2012). The remaining reads were then mapped on the hg19 (University of California, Santa Cruz) human genome with Tophat2 (Kim et al, 2013). 20.6–34.8 million reads (89.0–90.1%) were mapped. Read count analysis was performed with an R/Bioconductor package QuasR (Gaidatzis et al, 2015). A threshold of "fragments per kilobase of transcript per million mapped reads" (FPKM) greater than two was applied to select genes for subsequent differential gene expression analysis with an R/Bioconductor package edgeR (Robinson et al, 2010; McCarthy et al, 2012).

RT-qPCR was performed to determine transcript levels of selected transcripts in WT and *Helz*-null cells. Briefly, in the complementation assay described in Fig 6D, HEK293T *Helz*-null cells, plated in a six-well plate, were transfected with 1 and 4 μg of pT7-GFP-HELZ, as indicated. 48 h posttransfection, total RNA was extracted and reverse-transcribed using random hexamer primers. mRNA levels were subsequently determined by RT-qPCR using sequence-specific primers for the indicated transcripts and normalized to *gapdh* mRNA abundance in the same sample. qPCR primers were designed using Primer3 (Koressaar & Remm, 2007; Untergasser et al, 2012) or Primer-BLAST (Ye et al, 2012) and are listed in Table S5. Normalized expression ratios of the transcripts from three independent experiments were determined using the Livak method (Livak & Schmittgen, 2001).

## Immunofluorescence

HEK293T WT and *Ddx6*-null cells were grown on poly-D-lysine (Sigma-Aldrich)–coated cover slips. Cells were fixed with 4% paraformaldehyde for 10 min and permeabilized with 0.1% Triton X-100 in PBS (10 min). Staining with anti-DDX6 or anti-p70S6K (EDC4) antibodies was performed in PBS containing 10% FBS and 0.1% Tween 20 for 1 h. Alexa Fluor 594–labeled secondary antibody (Thermo Fisher Scientific) was used at 1:1,000 dilution. Nuclei were stained with Hoechst stain solution (Sigma-Aldrich). Cells were mounted using Fluoromount-G (Southern Biotech). The images were acquired using a confocal laser scanning microscope (Leica TCS SP8).

## Polysome profiling

Polysome profiles for HEK293T WT, *Helz*-null, and *Ddx6*-null cell lines were obtained as described before (Kuzuoglu-Ozturk et al, 2016).

## Data availability

Raw sequencing reads and the processed data files corresponding to read counts and normalized abundance measurements generated

in this study were deposited in the GEO under the accession number GSE135505.

## Supplementary Information

## Acknowledgements

We dedicate this paper to the memory of Elisa Izaurralde who sadly passed away before the conclusion of this work. Elisa was instrumental in the design and supervision of the study and in the drafting and correction of early versions of the manuscript. Due to journal policy, Elisa is not listed as an author. We want to thank Eugene Valkov for extensive language editing, Heike Budde for support in the RT-qPCR experiments, and Sigrun Helms for excellent technical assistance. We are grateful to all current and former members of the Izaurralde laboratory, who provided insightful comments on the manuscript. This work was supported by the Max Planck Society.

### Author Contribution

A Hanet: formal analysis, investigation, methodology, project administration, and writing—original draft, review, and editing.
F Räsch: investigation.
R Weber: formal analysis and investigation.
V Ruscica: investigation.
M Fauser: investigation.
T Raisch: resources.
D Kuzuoğlu-Öztürk: investigation.
C-T Chang: resources.
D Bhandari: investigation.
C Igreja: formal analysis, methodology, project administration, and writing—review and editing.
L Wohlbold: formal analysis, methodology, project administration, and writing—review and editing.

### Conflict of Interest Statement

The authors declare that they have no conflict of interest.

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
