## [Reviewer comments · Life Science Alliance]

Life Science Alliance

HELZ directly interacts with CCR4-NOT and causes decay of bound mRNAs

Aoife Hanet, Felix Raesch, Ramona Weber, Vincenzo Ruscica, Maria Fauser, Tobias Raisch, Duygu Kuzuoglu-Ozturk, Chung-Te Chang, Dipankar Bhandari, Catia Igreja, and Lara Wohlbold

DOI: <https://doi.org/10.26508/lsa.201900405>

Corresponding author(s): Catia Igreja, Max Planck Institute for Developmental Biology and Lara Wohlbold, Max Planck Institute for Developmental Biology

Review Timeline:

Submission Date:	2019-05-02
Editorial Decision:	2019-05-29
Revision Received:	2019-08-18
Editorial Decision:	2019-09-11
Revision Received:	2019-09-13
Accepted:	2019-09-13

Scientific Editor: Andrea Leibfried

Transaction Report:

May 29, 2019

Re: Life Science Alliance manuscript #LSA-2019-00405

Dr. Catia Igreja
Max Planck Institute for Developmental Biology
Biochemistry
Max-Planck-Ring 5
Tuebingen D-72076
Germany

Dear Dr. Igreja,

Thank you for submitting your manuscript entitled "HELZ directly interacts with CCR4-NOT and causes decay of bound mRNAs" to Life Science Alliance. The manuscript was assessed by expert reviewers, whose comments are appended to this letter.

As you will see, the reviewers appreciate the quality of the data provided. However, both reviewers note lack of insight into the potential physiological relevance of HELZ for deadenylation and reviewer #2 currently does not support publication of your work here. Reviewer #2 furthermore notes a few missing controls.

We discussed your work in light of these comments and think that the reviewers provide constructive input that will allow you to test for the physiological significance (such as via HELZ depletion). We would thus like to invite you to submit a revised version, addressing all comments of the reviewers. We do not expect you to identify the physiological role of HELZ entirely, but at least some demonstration of relevance would be needed.

Thank you for this interesting contribution to Life Science Alliance. We are looking forward to receiving your revised manuscript.

Sincerely,

B. MANUSCRIPT ORGANIZATION AND FORMATTING:

Reviewer #1 (Comments to the Authors (Required)):

Regulation of mRNA decay and translational control by RBPs has received significant attention in recent years. The role of the CCR4-NOT complex and its associated factors in this process are of particular importance. HELZ has been identified as a likely candidate involved in these processes by several studies. However, its precise role and mode of function have not been well described and thus this manuscript is timely and of significant interest to the RNA research community. Hanet et al, demonstrate that HELZ directly interacts with the CCR4-NOT complex and using reporter mRNAs show that HELZ promotes mRNA decay and translational repression.

Major comments:

1. While the data presented in the manuscript are quite compelling in demonstrating the interaction of HELZ with CCR4-NOT complex, elucidating the required motifs for the interactions, as well as its impact on the induction of mRNA decay and translational repression of the target mRNAs, there are some unanswered questions. It is not really clear whether the HELZ recruits CCR4-NOT complex to the mRNA or is associated with it and itself is recruited to the mRNA along with CCR4-NOT. Would depleting HELZ impact the CCR4-NOT-induced translational repression or mRNA decay?
2. In Figure S1C, the dmHELZ-N induces a significant increase in luciferase activity. Can the authors explain the reason for that and the discrepancy between the effect of human HELZ-N and dmHELZ-N?

Reviewer #2 (Comments to the Authors (Required)):

The authors report that an SF1 family helicase, HELZ, destabilizes a reporter mRNA to which it is tethered. HELZ induces deadenylation of the reporter: it associates with the CCR4-NOT complex, the complex is required for mRNA destabilization, and, when decapping is inhibited, the reporter accumulates in a deadenylated form. HELZ can also repress translation independently of deadenylation. This is dependent again on the CCR4-NOT complex and on the helicase DDX6.

The manuscript is technically very well done; I have few comments in this respect. The main deficit is that the authors merely demonstrate what HELZ CAN do (and probably does do in some unknown biological situation and without tethering), but a biological role or targets of HELZ are not identified.

Specific comments:

1. The authors observe a reduction in mRNA levels upon tethering of HELZ and conclude that the RNA is destabilized. Although an increase in decay rate is not directly shown, the conclusion is overall justified by the fact that the decrease in mRNA levels depends on mRNA decay enzymes: CCR4-NOT and the decapping complex. However, it would be appropriate for the authors to demonstrate, by the obvious RNaseH/oligo(dT) experiment, that the shortened mRNA accumulating upon inhibition of decapping is actually deadenylated.
2. I believe the experiments shown in Fig. 4 H-K are lacking the protein expression controls (HELZ

and GW182).

3. DDX6 knock-out cells: Do the cells have any obvious growth phenotype? I would not be surprised if the knock-out had an effect on general translation and perhaps mRNA levels. Please state if this is the case and how big the effect is compared to that of HELZ tethering. Please explain how exogenous GFP-DDX6 expression was done.

The authors state that the expression of HA-HELZ was similar in WT and DDX6-complemented cells (bottom of p. 12), and this is supported by the data in Fig. 5F. However, HA-HELZ appears to be overexpressed in the DDX6 null cells. Do the authors want to comment?

Minor points:

Abstract, last sentence: I agree that the data overall „hint at an important and conserved role for HELZ in the regulation of gene expression". However, the fact that repression and decay require the CCR4-NOT complex is not a good justification for this statement.

Introduction, p. 3, line 6: „.....either SF1 or SF2, which are characterized.... Are both superfamilies characterized by the tandem RecA-like domains?

four lines down: Insert „superfamily" behind „SF2".

p. 4, second paragraph, second to last line: „Collectively, these findings suggest...." In my opinion, only the very last finding reported in this paragraph (interaction with CCR4-NOT) suggests that HELZ has a role in regulating stability and translation of mRNA.

Results, p. 6, line 8: „identified reported" - delete one of the two.

p. 7, second paragraph, second line: „in vitro pulldown" - delete „in vitro" - pull-downs cannot be performed in vivo.

four lines down: Provide a reference and perhaps a brief explanation for GB1.

p. 11, second line: Provide a reference for Ge-1 and perhaps a brief explanation why it was depleted together with DCP1.

Response to reviewer 1

The referee states that *“Regulation of mRNA decay and translational control by RBPs has received significant attention in recent years. The role of the CCR4-NOT complex and its associated factors in this process are of particular importance. HELZ has been identified as a likely candidate involved in these processes by several studies. However, its precise role and mode of function have not been well described and thus this manuscript is timely and of significant interest to the RNA research community. Hanet et al, demonstrate that HELZ directly interacts with the CCR4-NOT complex and using reporter mRNAs show that HELZ promotes mRNA decay and translational repression.”*

We thank the reviewer for their positive comments.

The major comments of the reviewer are:

1. *“While the data presented in the manuscript are quite compelling in demonstrating the interaction of HELZ with CCR4-NOT complex, elucidating the required motifs for the interactions, as well as its impact on the induction of mRNA decay and translational repression of the target mRNAs, there are some unanswered questions. It is not really clear whether the HELZ recruits CCR4-NOT complex to the mRNA or is associated with it and itself is recruited to the mRNA along with CCR4-NOT. Would depleting HELZ impact the CCR4-NOT-induced*

translational repression or mRNA decay?”

To understand if HELZ has any impact on CCR4-NOT-induced mRNA decay or translational repression, we tethered human NOT1 to the R-Luc-6xMS2bs (to address mRNA decay) and to the R-Luc- 6xMS2bs-A₉₅-MALAT1 (to address translational repression) reporters in wild type cells or CRISPR-Cas9 engineered HELZ-null cells. Tethered NOT1 efficiently represses the expression of these R-Luc reporters in both cellular contexts. These results were included in Fig S3 and indicate that HELZ might only be required for the deadenylation and decay of a subset of mRNAs.

To further address this question, we have conducted transcriptome wide analysis of the HELZ-null cells via RNA-Seq. The data included in Figs 6, S4 and S5 of the revised manuscript shows HELZ depletion has a significant impact on gene expression with 1682 upregulated and 1830 downregulated transcripts (from a total of 10978 genes). Interestingly, mRNAs with increased levels in the absence of HELZ were overrepresented for genes encoding plasma membrane and extracellular proteins particularly involved in neurogenesis and nervous system development. This data supports the notion that HELZ might be required for the control of a specific subset of transcripts and opens the venue to further explore the involvement of HELZ, and its association with the CCR4-NOT complex, in the biological process of neurogenesis.

2. *“In Figure S1C, the dmHELZ-N induces a significant increase in luciferase activity. Can the authors explain the reason for that and the discrepancy between the effect of human HELZ-N and dmHELZ-N?”*

We agree with the reviewer that *Dm* HELZ-N induces a significant increase in luciferase activity. In the case of human HELZ-N (Figure 3A and B), we measured its effect only at the mRNA level, as the reporter used in these experiments contains only the β -globin ORF. We have tethered, however, the human HELZ fragments also to the R-Luc-6xMS2bs reporter. This data was not included in the manuscript but is now shown below for inspection by the reviewer. Once again, HELZ-N is inactive and HELZ-C mimics full length HELZ in the ability to repress and degrade the reporter mRNA. In contrast to *Dm* HELZ-N, the human counterpart does not change luciferase activity.

Figure 1. Tethering assay in HEK293T cells using the R-Luc-6xMS2bs reporter with MS2-HA-HELZ full length and N- and C-terminal fragments. A plasmid encoding F-Luc-GFP served as control. The graph shows the quantification of protein and mRNA levels of the R-Luc-6xMS2bs reporter normalized to the levels of the control reporter and set to 100 for MS2-HA. The mean values +/- SD are shown. A representative northern blot is depicted next to the graph.

The discrepancy between the two HELZ-N protein fragments could be associated with the fact that *Dm* HELZ-N (1211 residues) is longer than human HELZ-N (1050 residues) and contains non-conserved regions (particularly at the very N-terminal region; residues 1-329, Figure 1A and B of the manuscript). Although only speculative, the differences in sequence composition could provide additional and non-conserved functions and/or interactions to the *Dm* protein that are exacerbated or deregulated in the absence of HELZ-C.

Response to reviewer 2

Reviewer 2 states that:

“The authors report that an SF1 family helicase, HELZ, destabilizes a reporter mRNA to which it is tethered. HELZ induces deadenylation of the reporter: it associates with the CCR4-NOT complex, the complex is required for mRNA destabilization, and, when decapping is inhibited, the reporter accumulates in a deadenylated form. HELZ can also repress translation independently of deadenylation. This is dependent again on the CCR4-NOT complex and on the helicase DDX6.

The manuscript is technically very well done; I have few comments in this respect. The main deficit is that the authors merely demonstrate what HELZ CAN do (and probably does do in some unknown biological situation and without tethering), but a biological role or targets of HELZ are not identified.”

We thank the Reviewer for the positive comments.

To gain more insight into the potential biological role and targets of HELZ, we have generated, using CRISPR-Cas9 genome editing, HEK293T HELZ-null cells. The characterization of this cell line and the analysis of transcriptome wide changes have been included in the revised version of the manuscript. This data points out that HELZ depletion significantly impacts the expression of a specific subset of genes. In particular, upregulated mRNAs in the absence of HELZ are enriched in gene ontology categories representing membrane and extracellular proteins associated with the biological process of neurogenesis or nervous system development. Our results suggest that HELZ, and possibly its associating partners such as the CCR4-NOT complex, might have a function in the regulation of the expression and/or stability of specific transcripts during neurogenesis.

This reviewer also has specific comments:

1. *“The authors observe a reduction in mRNA levels upon tethering of HELZ and conclude that the RNA is destabilized. Although an increase in decay rate is not directly shown, the conclusion is overall justified by the fact that the decrease in mRNA levels depends on mRNA decay enzymes: CCR4-NOT and the decapping complex. However, it would be appropriate for the authors to demonstrate, by the obvious RNaseH/oligo(dT) experiment, that the shortened mRNA accumulating upon inhibition of decapping is actually deadenylated.”*

We have followed the reviewers' suggestion and have added the corresponding experiment to the revised manuscript (Fig S1A).

2. *“I believe the experiments shown in Fig. 4 H-K are lacking the protein expression controls (HELZ and GW182).”*

We agree with the reviewer and have added the protein expression control for the experiment shown in Fig. 4. For consistency reasons, we have also added to the revised figures the protein expression controls for the experiments shown in Figures 3 and 5A-C (data not shown in the original version of the manuscript).

3. *“DDX6 knock-out cells: Do the cells have any obvious growth phenotype? I would not be surprised if the knock-out had an effect on general translation and perhaps mRNA levels.*

Please state if this is the case and how big the effect is compared to that of HELZ tethering.

We agree with the reviewer that depletion of a major regulator of gene expression such as DDX6 may introduce global cellular changes. To exclude any impact on the results observed with the tethering system caused by loss of DDX6, such as interfering with reporter and proteins expression levels, we provide in the revised version of the manuscript a thorough characterization of the DDX6-null cell line (Fig. S2).

Importantly, the DDX6 knock-out cells that we generated do not display an obvious growth phenotype in the cultured conditions that we have used (DMEM supplemented with 10% foetal bovine serum, glutamine, penicillin and streptomycin). We also dismissed effects on general translation caused by DDX6-depletion using polysome profiling analysis. The DDX6-null cells did however show an abnormal distribution of the P-body marker EDC4 (Enhancer of Decapping 4; Fig. S2C and D), confirming the role of DDX6 in the formation and/or maintenance of P-bodies [as observed in other DDX6 knock-out cellular models; (Lumb et al. 2017; Freimer et al. 2018)].

We also observed that repression of R-Luc activity by the silencing domain of the miRISC component TNRC6A (TNRC6A-SD) was not affected in DDX6-null cells (Fig. S2E and F). Altogether, these results support the notion that reduced translational repressor function of HELZ in DDX6-null cells is specific and not a consequence of general cellular changes ongoing in the knock-out cells.

Please explain how exogenous GFP-DDX6 expression was done.”

The conditions of the complementation assay performed in the DDX6-null cells are detailed in the materials and methods.

“The authors state that the expression of HA-HELZ was similar in WT and DDX6-complemented cells (bottom of p. 12), and this is supported by the data in Fig. 5F. However, HA-HELZ appears to be overexpressed in the DDX6 null cells. Do the authors want to comment?”

In Fig. 5F, the appropriate transfection control was missing. We have repeated the experiment (revised Fig. 5G and H) and analysed the expression of HA-HELZ in WT and

DDX6-null cells by western blot using F-Luc-GFP as a transfection control. We observed that the expression of HA-HELZ in WT and DDX6-null cells is mostly similar, and the observed difference in the original Fig. 5F was most probably resulting from differences in transfection efficiency between the experimental conditions. However, we observed that overexpression of GFP-DDX6 appeared to reduce the expression of the transfection control F-Luc-GFP and also HA-HELZ, even if expressed at levels equivalent to endogenous DDX6. We assume that overexpressed GFP-DDX6 might have some inhibitory effect on gene expression. Nevertheless, the repression of the R-Luc reporter by HELZ in DDX6-null cells was still restored in these conditions (overexpression of GFP-DDX6).

This reviewer also has minor comments:

1. *“Abstract, last sentence: I agree that the data overall „hint at an important and conserved role for HELZ in the regulation of gene expression”. However, the fact that repression and decay require the CCR4-NOT complex is not a good justification for this statement.”*

We have rephrased the last sentence of the abstract in the revised version of the manuscript.

2. *“Introduction, p. 3, line 6: „.....either SF1 or SF2, which are characterized.... Are both superfamilies characterized by the tandem RecA-like domains?”*

Yes, both SF1 and SF2 superfamilies are characterized by tandem RecA-like domains.

3. *“four lines down: Insert „superfamily” behind „SF2”.”*

We have edited the section accordingly.

4. *“p. 4, second paragraph, second to last line: „Collectively, these findings suggest....” In my opinion, only the very last finding reported in this paragraph (interaction with CCR4-NOT) suggests that HELZ has a role in regulating stability and translation of mRNA.”*

We thank the reviewer for pointing out this inconsistency. It is now corrected in the revised

version of the manuscript.

5. *“Results, p. 6, line 8: „identified reported” - delete one of the two.”*

We thank the reviewer for pointing out this inconsistency. It is now corrected in the revised version of the manuscript.

6. *“p. 7, second paragraph, second line: „in vitro pulldown” - delete „in vitro” - pull-downs cannot be performed in vivo.”*

We thank the reviewer for pointing out this inconsistency. It is now corrected in the revised version of the manuscript.

7. *“four lines down: Provide a reference and perhaps a brief explanation for GB1.”*

The information regarding the GB1 domain was only provided in the Materials and Methods section. A brief explanation has been added to the revised Results section.

8. *“p. 11, second line: Provide a reference for Ge-1 and perhaps a brief explanation why it was depleted together with DCP1.”*

The information has been added to the results section of the revised manuscript.

References:

Freimer JW, Hu TJ, Blalock R. 2018. Decoupling the impact of microRNAs on translational repression versus RNA degradation in embryonic stem cells. *Elife* 7.

Lumb JH, Li Q, Popov LM, Ding S, Keith MT, Merrill BD, Greenberg HB, Li JB, Carette JE.
2017. DDX6 Represses Aberrant Activation of Interferon-Stimulated Genes. *Cell Rep*
20: 819-831.

September 11, 2019

RE: Life Science Alliance Manuscript #LSA-2019-00405R

Dr. Catia Igreja
Max Planck Institute for Developmental Biology
Biochemistry
Max-Planck-Ring 5
Tuebingen D-72076
Germany

Dear Dr. Igreja,

Thank you for submitting your revised manuscript entitled "HELZ directly interacts with CCR4-NOT and causes decay of bound mRNAs". As you will see, both reviewers appreciate the introduced changes, and we would thus be happy to publish your paper in Life Science Alliance. Before sending you an official acceptance letter, please:

- address the comments made by reviewer #2
- I think it is not necessary to include 'However, due to journal policy we were unable to include her as an author.' in the acknowledgement section, please remove.

A. FINAL FILES:

-- Summary blurb (enter in submission system): A short text summarizing in a single sentence the study (max. 200 characters including spaces). This text is used in conjunction with the titles of papers, hence should be informative and complementary to the title. It should describe the context

and significance of the findings for a general readership; it should be written in the present tense and refer to the work in the third person. Author names should not be mentioned.

B. MANUSCRIPT ORGANIZATION AND FORMATTING:

Sincerely,

Andrea Leibfried, PhD
Executive Editor
Life Science Alliance
Meyershofstr. 1
69117 Heidelberg, Germany
t +49 6221 8891 502
e a.leibfried@life-science-alliance.org
www.life-science-alliance.org

Reviewer #1 (Comments to the Authors (Required)):

The authors have satisfactorily addressed my comments. I recommend publication.

Reviewer #2 (Comments to the Authors (Required)):

This is a revised manuscript.

I am satisfied with the authors' response to my criticism and support publication.

Two minor issues to be addressed:

Fig. S4 appeared to be incomplete.

p. 18, second paragraph: The sentence 'Thus, DDX6 contributes...' is poorly formulated; please rephrase.

Response to reviewer 2

The referee states that “This is a revised manuscript. I am satisfied with the authors' response to my criticism and support publication.”

We thank the reviewer for their positive comments.

The minor comments of the reviewer are:

1. *“Fig. S4 appeared to be incomplete.”*

Fig. S4 contains three panels (A-C). The complete Fig. S4 has been uploaded to the LSA server. We apologize to the reviewer if the figure was incomplete.

2. *“p.18, second paragraph: The sentence 'Thus, DDX6 contributes...' is poorly formulated; please rephrase.”*

We have rephrased the sentence in the revised version of the manuscript.

September 13, 2019

RE: Life Science Alliance Manuscript #LSA-2019-00405RR

Dr. Catia Igreja
Max Planck Institute for Developmental Biology
Biochemistry
Max-Planck-Ring 5
Tuebingen D-72076
Germany

Dear Dr. Igreja,

Thank you for submitting your Research Article entitled "HELZ directly interacts with CCR4-NOT and causes decay of bound mRNAs". It is a pleasure to let you know that your manuscript is now accepted for publication in Life Science Alliance. Congratulations on this interesting work.

DISTRIBUTION OF MATERIALS:

Again, congratulations on a very nice paper. I hope you found the review process to be constructive and are pleased with how the manuscript was handled editorially. We look forward to future exciting submissions from your lab.

Sincerely,
